# Microwave Heating Healing of Asphalt Mixture with Coal Gangue Powder and Basalt Aggregate

Bin Zhang [1], Xu Gao [1], Shi Xu [1,2,3,*], Xinkui Yang [4], Qin Tian [1] and Jiayi Liu [5]

1  School of Civil Engineering and Architecture, Wuhan University of Technology, Wuhan 430070, China; 314258@whut.edu.cn (B.Z.); x.gao@whut.edu.cn (X.G.); 334166@whut.edu.cn (Q.T.)
2  Faculty of Civil Engineering and Geosciences, Delft University of Technology, 2628 CN Delft, The Netherlands
3  Hubei Key Laboratory of Roadway Bridge and Structure Engineering, Wuhan University of Technology, Wuhan 430070, China
4  State Key Laboratory of Silicate Materials for Architectures, Wuhan University of Technology, Wuhan 430070, China; yangxk@whut.edu.cn
5  School of Materials Science and Engineering, Wuhan University of Technology, Wuhan 430070, China; 322073@whut.edu.cn
*  Correspondence: s.xu-1@tudelft.nl

**Abstract:** Microwave heating is an effective method to achieve autonomic crack healing in asphalt mixtures, and the use of microwave-absorbing materials can largely improve this healing efficiency. As a solid waste, coal gangue contains metal oxides, which shows the possibility of microwave heating. In order to further promote the application of coal gangue in the microwave healing of asphalt mixtures, this study looks into the synergistic effect of basalt and coal gangue powder (CGP) on the microwave heating self-healing of an asphalt mixture. The mechanical performance, water stability, low-temperature crack resistance and microwave healing efficiency of the asphalt mixture were investigated using the immersion Marshall test, standard Marshall test, Cantabro test and semi-circular bending (SCB), and healing tests, respectively. The results indicated that the addition of CGP in asphalt mixture can improve the microwave heating speed, which also showed a significant advantage in water stability and fracture energy recovery. The research results will further promote the utilization rate of coal gangue.

**Keywords:** self-healing asphalt; coal gangue powder; basalt; microwave healing; pavement performance; healing efficiency

## 1. Introduction

Coal plays a major role in the supplement of world energy [1]. Considering coal as a major energy source [2], China has become the world's largest producer and consumer of coal [3–5]. The process of coal mining and washing produces a significant amount of coal gangue [6–8]. Currently, the amount of accumulated coal gangue in China exceeds 7 billion tons [9], which not only occupies a large area of land, but also leads to serious environmental pollution [10–13]. Therefore, it is urgent to expand the resource utilization of coal gangue.

Some studies have found that coal gangue powder (CGP) has the potential to be used as a filler in asphalt mixtures [14–16]. Liu et al. [17] investigated the asphalt modification effect with different types of CGP. The results showed that these CGPs can enhance the performance of asphalt, especially its high temperature performance. Amir et al. [18,19] looked into the applicability of CGP as a replacement of conventional limestone fillers in asphalt mixtures. The results indicated that the addition of CGP enhanced the elastic modulus, average fatigue lives and water stability properties of asphalt mixtures. Hong et al. [20] found that the use of CGP to replace mineral powder can improve the low-temperature crack resistance of asphalt mixtures.

Temperature and fatigue cracks are two of the main distresses that seriously threaten the durability of asphalt pavements [21]. Through artificial heating approaches, heating the asphalt mixture to sufficient temperature thresholds, the cracks tend to be closed by asphalt molecules constantly infiltrating and dispersing, which can extend the service life of asphalt pavements [22]. With the advantages of rapid heating, energy saving and low cost, microwave heating technology has been widely used as an asphalt crack heating-healing mechanism in road engineering [22,23]. Juan Gallgo et al. [24] proved that compared with other thermal induction methods (electromagnetic induction and infrared), microwave heating technology had a better repair effect on asphalt pavement damage because of the lower energy consumption and simple equipment. Various additives containing wave-absorbing materials were reported to effectively improve the microwave heating healing for asphalt mixtures, such as steel slag, steel velvet, ferrite, carbon black and fly ash [23,25–29]. Li et al. [25] found that the addition of steel slag filler can increase the heat release of asphalt mastic under microwave irradiation, thus improving the self-healing performance. Zhu et al. [23] concluded that the asphalt mixture filled with ferrite had the most effective absorption of microwaves, thus expediting the spread of asphalt molecules across the crack surface and improving the healing property. The wave-absorbing materials can be divided into electric-loss microwave absorbing materials (pyrolytic carbon black, Si, etc.) and magnetic-loss microwave absorbing materials (ferrite, steel slag, etc.) [30–34]. Wang et al. [35] investigated the microwave self-healing property and heating mechanism of the asphalt mixture with limestone and basalt aggregates. The results indicated that basalt can enhance the microwave heating self-healing performance of asphalt mixtures because it contains elements such as Si, Fe and Al. Coal gangue mainly contains Si, Al, Fe, Ca and Mg oxides and several rare metals and can be heated by microwaves. Li et al. [36] prepared two types of CGP as fillers to replace the conventional limestone fillers in different proportions in the asphalt mixture and investigated the low-temperature cracking resistance property and microwave heating healing efficiency. The results show that CGP can enhance the low-temperature cracking resistance and microwave self-healing efficiency of asphalt mixtures. Compared to metal materials such as steel fiber, CGP and basalt were mineral materials, whose addition did not change the composition of the asphalt mixture and thus will not cause the corrosion issue. Thus, CGP and basalt are more suitable for asphalt pavements due to the better integration with asphalt mixtures.

Both CGP and basalt have the possibility of being heated by microwaves. Basalt aggregate has high strength and resistance to slip and wear, but its adhesion to asphalt is somewhat weak [37], while CGP has a larger specific surface area and a rough shape [38,39], so its open pores can absorb asphalt. CGP has the potential to be utilized to enhance the adhesion between basalt and asphalt, so as to improve the water stability of asphalt and mixture.

Therefore, the main purpose of this research is to investigate the effect and feasibility of the combined effect of CGP and basalt on improving the pavement performance of asphalt mixtures and self-healing property of microwave heating. CGP is used to replace limestone filler to improve the adhesion between basalt and asphalt and promote the application of coal gangue in the microwave heating self-healing of asphalt mixture. This research used an X-ray fluorescence spectrometer (XRF) to analyze the chemical composition. The temperature distribution of asphalt mixtures with different CGP replacements was investigated using an infrared thermal imager. The low-temperature crack resistance and microwave heating self-healing properties of the asphalt mixture with different proportions of CGP and basalt were evaluated using an SCB test through the process of fracture healing. Using the Marshall test and the Cantabro test, the effect of CGP on improving the water stability of asphalt mixture was evaluated. The research outline of this study is shown in Figure 1. The results of this experiment will be beneficial to further stimulate the recycling of coal gangue.

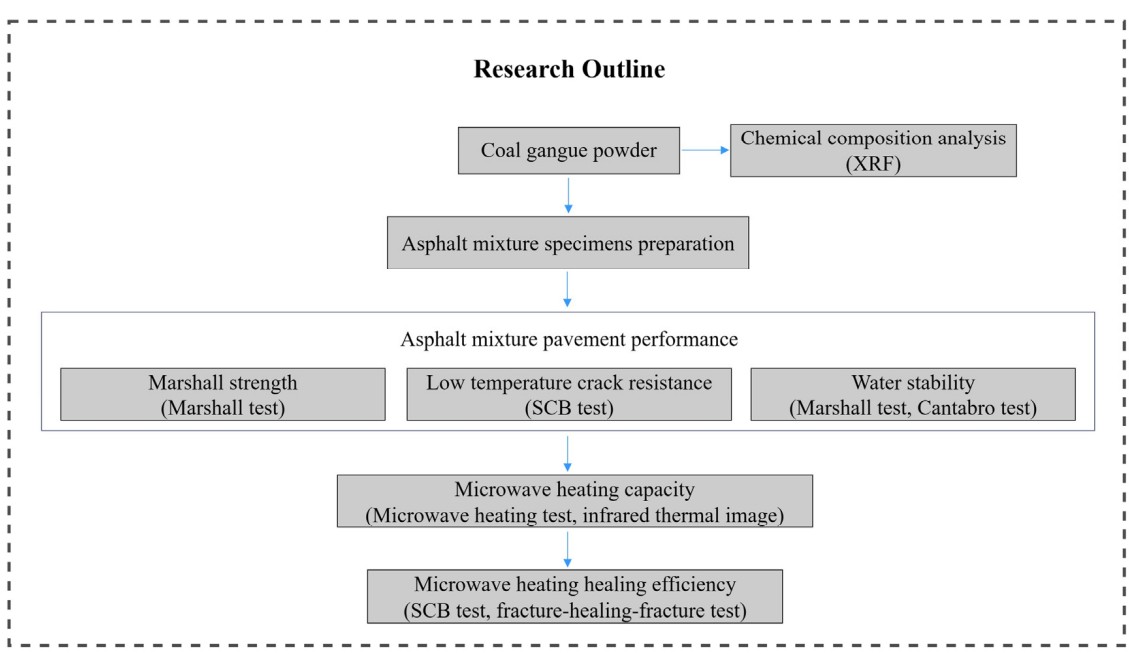

**Figure 1.** Flow-process scheme for this study.

## 2. Materials and Methods

### 2.1. Asphalt and Aggregate

This experiment used the PEN 70 base asphalt from the Hubei Guochuang Road Material Technology Co., Ltd. (Wuhan, China). Its basic technical specifications were tested in accordance with the Chinese standard test methods of bitumen and bituminous mixtures for highway engineering (JTG E20-2011), which are listed in Table 1. Originating from Jingshan, Hubei Province, basalt aggregates in accordance with AC-13 grade were chosen to prepare the asphalt mixture. Its physical properties are listed in Table 2, which were tested in accordance with the Chinese test methods of aggregate for highway engineering (JTG E42-2005). The various technical indicators tested for asphalt and basalt met Chinese technical specifications for construction of highway asphalt pavements (JTG F40-2004).

**Table 1.** Properties of PEN 70 base asphalt.

| Item | Values | Test Method | Criteria |
|---|---|---|---|
| Penetration (25 °C, 0.1 mm) | 69.8 | T 0604 | 60~80 |
| Ductility (15 °C, cm) | >100 | T 0605 | >100 |
| Softening point (°C) | 49.5 | T 0606 | $\geq$46 |

**Table 2.** Properties of the basalt.

| Item | Values | Test Method | Criteria |
|---|---|---|---|
| Crushing value (%) | 15.8 | T 0316 | $\leq$26 |
| Water absorption (%) | 0.58 | T 0304 | $\leq$2.0 |
| Apparent specific gravity (g/cm$^3$) | 2.913 | T 0304 | $\geq$2.60 |
| Los Angeles wear loss (%) | 19.8 | T 0317 | $\leq$28 |

### 2.2. Fillers

The two different types of CGP, one originating from Taiyuan, Shanxi, and one originating from Zhangjiakou, Hebei, were used as the replacement for conventional mineral powder. Conventional mineral powder (CMP) is limestone powder, which was used in reference samples. The replacement rate of the filler is listed in Table 3, the physical performance of the filler is shown in Table 4, and the appearance form of the filler is shown in

Figure 2. It can be seen from Table 4 that the average particle size and hydrophilicity coefficient of both kinds of CGP are smaller than those of the CMP, while their specific surface area is larger than that of the CMP. In particular, CGP2 has the smallest average particle size of 20,260 nm and the largest specific surface area of 0.49 $m^2/g$, which is conducive to increasing the contact area between the filler and asphalt.

**Table 3.** The proportion of various fillers in asphalt mixture.

| Serial Number | Proportion | Types |
|---|---|---|
| 1 | 100% CMP | CMP-100% |
| 2 | 50% CGP1 + 50% CMP | CGP1-50% |
| 3 | 50% CGP2 + 50% CMP | CGP2-50% |
| 4 | 100% CGP1 | CGP1-100% |
| 5 | 100% CGP2 | CGP2-100% |

**Table 4.** The physical properties of various fillers in asphalt mixture.

| Types | Average Particle Size (nm) | Specific Surface Area ($m^2/g$) | Hydrophilicity Coefficient |
|---|---|---|---|
| CGP1 | 27,730 | 0.44 | 0.75 |
| CGP2 | 20,260 | 0.49 | 0.73 |
| CMP | 57,955 | 0.37 | 0.85 |

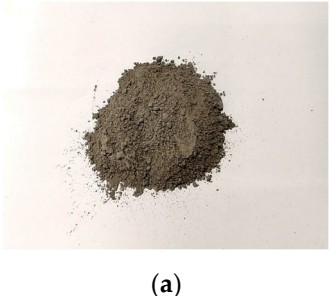
(**a**)

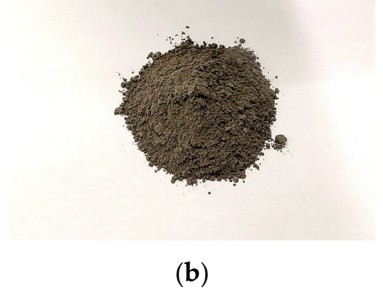
(**b**)

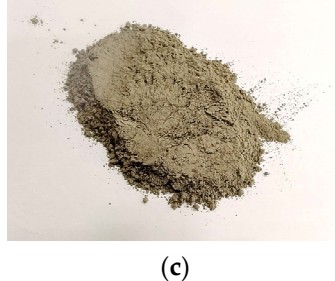
(**c**)

**Figure 2.** The appearance of various fillers: (**a**) CGP1, (**b**) CGP2, and (**c**) CMP.

*2.3. Asphalt Mixture Proportioning Design*

This study followed the AC-13 whose gradation curves are shown in Figure 3. The samples of the asphalt mixtures used in the tests were prepared according to the Chinese standard test methods of bitumen and bituminous mixtures for highway engineering (JTG E20-2011). The optimum asphalt content was determined using the Marshall method.

*2.4. Experimental Methods*

2.4.1. Chemical Composition Analysis

The main chemical compositions of different gangue powders were obtained by XRF detection (Type Zetium, Power 4 kW, analytical element range O8~Am95), so as to analyze their influence on the pavement properties and microwave heating self-healing performance of asphalt mixtures.

2.4.2. Stability of Asphalt Mixtures

The Marshall stability test was used to characterize the effect of CGP on the Marshall stability and water stability of asphalt mixture. The size of the standard Marshall specimen shall be 101.6 mm ± 0.2 mm in diameter and 63.5 mm ± 1.3 mm in height. The Marshall stability tests were carried out according to T0709 in JTG E20-2011. The SYD-0709A Marshall Stability Tester, made in Shanghai Changji Geological Instrument Co., Ltd., Shanghai, China, was used to carry out the immersion Marshall test and standard Marshall test [40,41]. The

standard Marshall test kept the specimens in a thermostatic temperature water basin at 60 °C for 30 min, while the immersion Marshall test required 48 h. After that, the specimens were taken out and placed on the Marshall stability tester to obtain the standard Marshall stability and immersion Marshall stability of the specimens. The Marshall test apparatus is shown in Figure 4. Residual Marshall stability (RMS) can be used to characterize the water stability of asphalt mixture, which was calculated by the following equation:

$$RMS = \frac{MS_1}{MS} \times 100 \tag{1}$$

where,

$RMS$ is the residual Marshall stability of specimens (%),
$MS$ is the standard Marshall stability of specimens (kN), and
$MS_1$ is the immersion Marshall stability of specimens (kN)

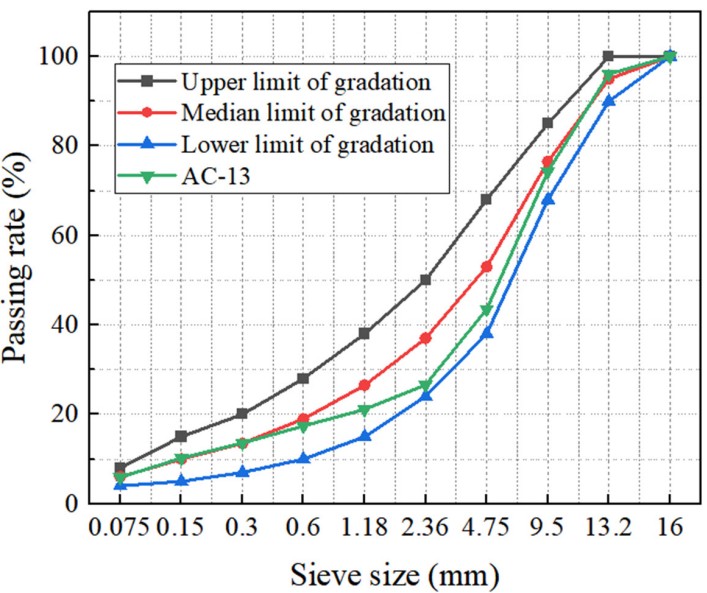

**Figure 3.** AC-13 aggregate grading curves.

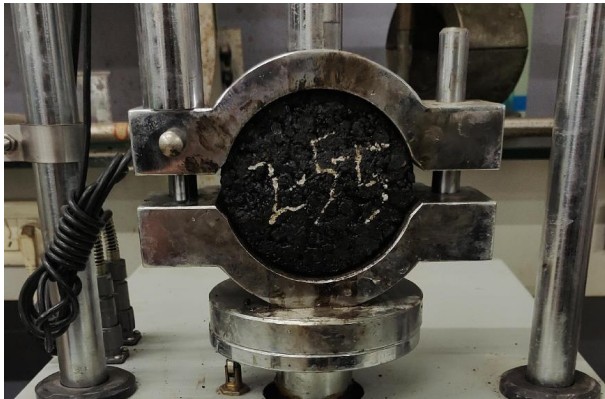

**Figure 4.** The standard Marshall samples under testing.

### 2.4.3. Cantabro Test

The effects of CGP on the adhesion property and water stability of asphalt mixture were analyzed by the water immersion and flying test on Marshall specimens, which was carried out according to T0733 in JTG E20-2011. The standard Marshall samples were taken out after standing in a thermostatic temperature water basin at 60 °C for 48 h and then

stood at room temperature for 24 h. The quality of scattered materials of asphalt mixture specimens was tested after 300 rotations of impact in the Los Angeles testing machine at 30 r/min. The calculation of the scattered losses named $\Delta S$ of the asphalt mixture based on the mass of the sample before and after testing, which was illustrated by the following equation:

$$\Delta S = \frac{m_0 - m_1}{m_0} \times 100 \tag{2}$$

where,

$\Delta S$ is the immersion scattered losses of the asphalt mixture (%),
$m_0$ is the mass of specimen before Cantabro testing (g), and
$m_1$ is the residual mass of specimen after Cantabro testing (g)

### 2.4.4. Cracking Resistance of Asphalt Mixes

The sample was cut into four semi-circular sheets with a diameter of about 101 mm and a thickness of about 32 mm by a cutting machine. Then, an intermediate notch with a width of 3mm and a height of 10 mm is cut in the center of each semicircle [24]. The samples were first stored at 0 °C for 4 h and then subjected to semi-circular bending tests with the UTM-25 Servo-Hydraulic Asphalt Mixtures Dynamic Test System to obtain the load displacement curves. In particular, the distance between the two pivot points on the test instrument was 80 mm, and the loading rate was 0.5 mm/min. Semi-circular specimens are shown in Figure 5. Peak stress can be used to evaluate the low-temperature crack resistance of asphalt mixture, the higher the value, the better the fracture resistance [28]. In this study, the fracture energy was used to quantify the fracture resistance of asphalt mixture and evaluation of the healing efficiency [35,42]. The fracture energy is calculated based on the fracture power of the sample and the effective area of the cross section, which was illustrated by the following equation:

$$G_f = \frac{W_f}{A_{lig}} \tag{3}$$

where,

$G_f$ is the fracture energy of the asphalt mixture (J·m$^{-2}$),
$W_f$ is the fracture power of the specimen (J), and
$A_{lig}$ is the effective area of the cross section of the specimen (m$^2$)

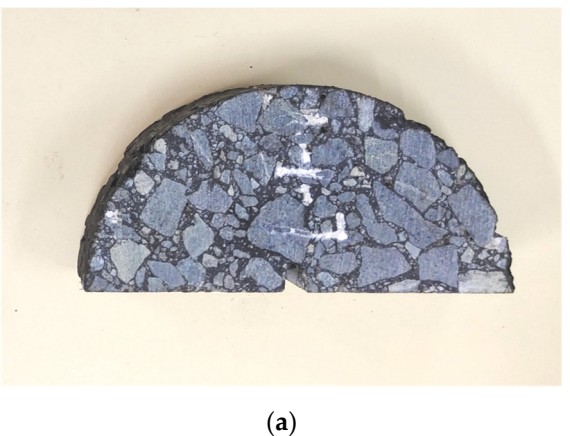
(a)

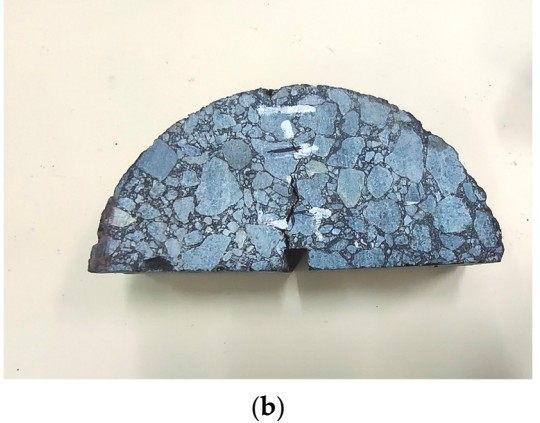
(b)

**Figure 5.** The SCB specimens: (**a**) before bending test and (**b**) after bending test.

### 2.4.5. Microwave Healing Test

In this study, the NN-GT353M microwave oven, with a maximum output power of 800 W and a working frequency of 2.45 GHz, produced by Panasonic of Japan, was used to heat the sample. The T420 infrared thermal imager, produced by the FLIR company in the United States, was used to observe the temperature distribution of the semi-circular specimen before fracture to determine the optimum heating period. First, the optimal heating period was determined by monitoring the surface temperature changes with microwave heating time, which is the minimum time that increases the surface temperature of asphalt mixture beyond the healing temperature 85 °C [22,43]. Figure 6 is an example of infrared thermal image of asphalt mixture under microwave heating.

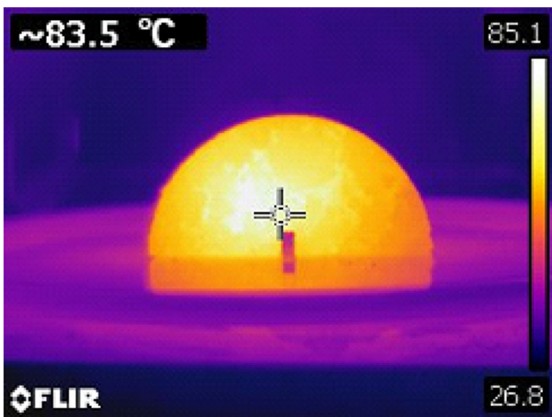

**Figure 6.** The infrared image example figure of asphalt mixture.

The fractured sample was dried for 8 h to take out the frost on the surface and inside of the sample at room temperature. The dried samples were then heated in the microwave oven to the optimum heating time, and then stored at room temperature for 8 h. The above two steps were repeated until the 6th bending test was completed. The calculation of the microwave heating healing rate named $R_x$ of the asphalt mixture was based on the fracture energy acquired from each semi-circular bending test, which was illustrated by the following equation:

$$R_x = \frac{G_{fx}}{G_{f1}} \times 100 \tag{4}$$

where,

$R_x$ is the healing efficiency of each cycle of asphalt mixtures (%),

$G_{f1}$ is the fracture energy of the first fracture test of asphalt mixtures (J·m$^{-2}$), and

$G_{fx}$ is the fracture energy acquired at the $x$ testing cycle, $1 \leq x \leq 6$ (J·m$^{-2}$)

## 3. Results and Discussion

### 3.1. Chemical Composition of Fillers

Table 5 shows the main chemical composition of the XRF for three types of fillers. The main components of the two CGP fillers are silicon oxide and metal oxides such as alumina and iron oxide. The combined content of these three oxides in CGP1 is 92.63%, which is higher than 85.12% in CGP2. The main component of the CMP is CaO, and its silicon, aluminum and iron oxides combined content is only 1.45%. Some studies have shown that Si is an electric loss microwave absorbing material [32], and ferrite can absorb microwaves by means of both dielectric loss and magnetic loss, mainly through magnetic loss [33]. The compounds composed of Si and related metals such as Fe and Al have good electromagnetic properties and can absorb microwaves [44,45]. Compared with CMP, CGP may have better microwave absorption efficiency. Therefore, the use of CGP instead of CMP is helpful to realize the rapid microwave heating ability of the asphalt mixture.

**Table 5.** Chemical composition of CGP1, CGP2 and CMP.

| Composition | SiO$_2$ | CaO | MgO | Al$_2$O$_3$ | Fe$_2$O$_3$ | K$_2$O | Na$_2$O |
|---|---|---|---|---|---|---|---|
| CGP1 | 57.74 | 0.20 | 1.04 | 30.58 | 4.31 | 2.76 | 1.10 |
| CGP2 | 53.72 | 2.51 | 0.87 | 26.39 | 5.01 | 2.44 | 1.13 |
| CMP | 0.72 | 52.86 | 2.27 | 0.61 | 0.12 | 0.35 | - |

### 3.2. Stability of the Asphalt Mixtures

Compared to other groups, the CMP-100% has the highest standard Marshall strength of 19.30 kN, from Figure 7. The standard Marshall [40] strength of CGP1-50% and CGP2-50% is 15.10 kN and 14.92 kN, respectively. The standard Marshall strength of CGP1-100% and CGP2-100% is 14.57 kN and 14.43 kN, respectively. It can be seen that the addition of the CGP reduces the standard Marshall strength of the asphalt mixture. According to the technical specifications for the construction of highway asphalt pavements (JTG F40-2004), the standard Marshall strength of asphalt mixtures for expressway or first-class highways is required to be at least 8 kN. From Figure 7, with the addition of CGP, the standard Marshall strength of various asphalt mixes gradually decreases, but is sufficient to meet the use requirements of expressway or first-class highways. When the content of CGP increases from 0 to 50%, the Marshall strength decreases by 21.8% (CGP1) and 22.7%(CGP2), respectively. When the content of CGP increases from 50% to 100%, the Marshall strength only decreases by 3.5% (CGP1) and 3.3% (CGP2), respectively. The two kinds of CGP have similar effects on the Marshall stability of asphalt mixtures. This shows that a higher substitution rate (50–100%) of coal gangue will not further affect the strength of asphalt mixtures. It is beneficial to improve the utilization rate of coal gangue.

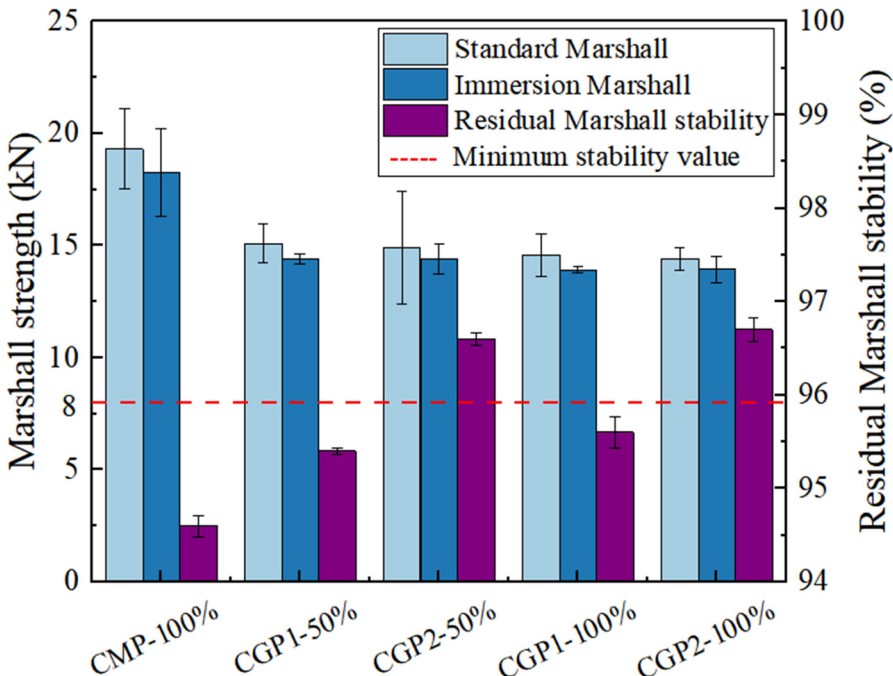

**Figure 7.** The standard Marshall strength, immersion Marshall strength and immersion loss rate of various asphalt mixes.

Based on the standard Marshall test, the CMP-100% has the highest water immersion Marshall strength of 18.27 kN, from Figure 7. The water immersion Marshall strength of CGP1-50% and CGP2-50% is 14.4 kN and 14.42 kN, respectively. The water immersion Marshall strength of CGP1-100% and CGP2-100% is 13.93 kN and 13.95 kN, respectively. The characteristics of the two kinds of CGP in the immersion Marshall test were similar,



and the immersion Marshall strength of them were basically the same. It can be seen that with the addition of CGP, the immersion Marshall strength gradually decreased, and the reduction rate gradually slowed down, which had the same trend as the standard Marshall strength.

From Figure 7, the residual Marshall stability of CMP-100% is 94.6%. The residual Marshall stability of CGP1-50% and CGP2-50% are 95.4% and 96.6%, respectively. The residual Marshall stability of CGP1-100% and CGP2-100% are 95.6% and 96.7%, respectively. The residual Marshall stability of each group meets the requirement of ≥80% proposed in JTG F40-2004. The addition of CGP can improve the residual Marshall stability of the asphalt mixture, and the effect of CGP2 is greater. Therefore, Marshall test results show that CGP reduces the strength, but improves water stability of the asphalt mixture.

This might be due to the grade ratio of CGP being finer compared with ordinary mineral powder, and the specific surface area of the material is increased, which means that the filler can interact with the asphalt and aggregate on a larger contact area, thus improving the water stability [39].

*3.3. Cantabro Test*

Figure 8 illustrates the scattered losses of asphalt mixtures with different fillers in the immersion Cantabro test. The scattered losses of all asphalt mixtures were less than 20%, which met the requirements of technical specification for the construction of highway asphalt pavements (JTG F40-2004). Among all these asphalt mixes, CMP-100% had the highest scattered losses rate of 5.5%. CGP1-50% had the scattered losses rate of 4.8%, which was reduced by 12.7% compared with the CMP-100%, and CGP1-100% had the lowest scattered losses rate of 4.6%, which was reduced by 16.4% compared with the CMP-100%. A smaller ΔS generally indicates a better adhesive property and the water stability for the asphalt mixture. It is also shown in Figure 8 that the incorporation of CGP can reduce the immersion scattered losses of the asphalt mixtures and improve the ability to resist water damage. And the results are better with 100% CGP content. This might be due to the fact that CGP has a larger specific surface area compared with CMP (Table 4), which increases the interaction area of asphalt aggregates and enhances the adhesion of asphalt aggregates, and CGP1 is more effective.

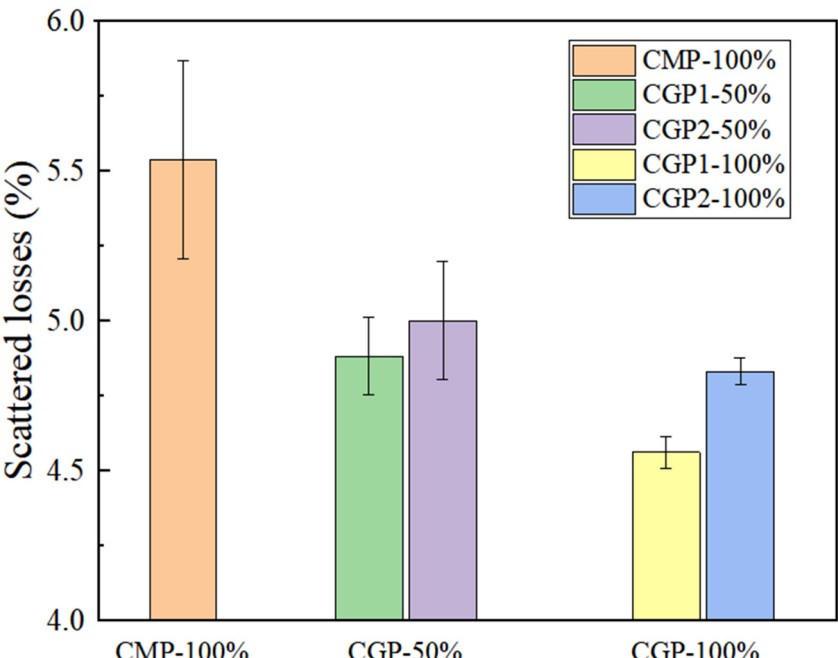

**Figure 8.** Scattered losses of asphalt mixtures with different fillers.

Combined with the results of the Marshall test and Cantabro test, it can be seen that CGP1 and CGP2 have similar effects on improving the water stability of the asphalt mixture, but different characterization methods have different effects. CGP1 is better at improving the scattered losses of asphalt mixtures, while CGP2 is better at improving the residual Marshall stability of asphalt mixtures. It is also found that an increasing amount of CGP contributes to a greater water stability improvement of the asphalt mixture.

### 3.4. Low Temperature Fracture Properties

Figure 9 presents the peak stresses acquired from the semi-circular bending tests. It can be seen that CMP-100% had the lowest peak stress of 2.9 kN and CGP1-50% had the highest peak stress of 4.3 kN, which was 1.48 times that of CMP-100%. The peak stress of CGP2-50%, CGP1-100% and CGP2-100% were 4.0 kN, 3.8 kN and 3.6 kN, respectively. All of them were higher than the control group. As shown in Figure 9, the addition of CGP was able to improve the low temperature fracture resistance of the asphalt mixtures. The reason for this phenomenon can be explained by the fact that the CGP has a larger surface area and can adsorb more asphalt to enhance the adhesion capacity. At 50% CGP content, the combined effect with mineral powder is more effective in improving low-temperature crack resistance. This suggests that the low-temperature fracture resistance of asphalt mixtures does not always improve with the addition of CGP. This may be due to the fact that excessive CGP will have a negative impact on low-temperature crack resistance because of its agglomeration and the poor penetration of asphalt [24]. This indicates that the CGP can improve the low-temperature cracking resistance of the asphalt mixture, and the effect is greater when the appropriate dosage (about 50%) is used.

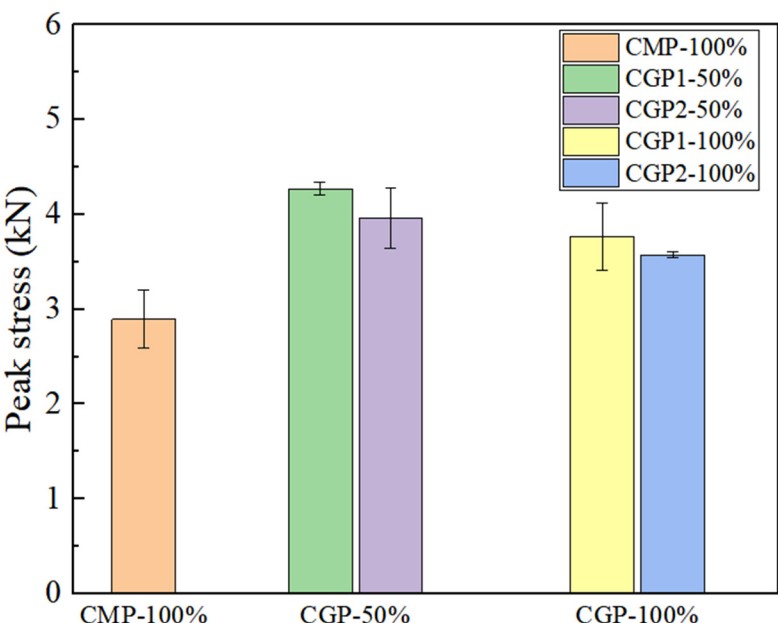

**Figure 9.** Peak stress at low-temperature fracture of various asphalt mixtures.

### 3.5. Microwave Heating Ability of Asphalt Mixes with Different Filler

Figure 10 shows the maximum surface temperatures of the five types of samples at different heating times, thus further evaluating the microwave heating capability of the mixtures with different fillers. When the heating time was 20 s, 30 s and 40 s, the average temperature of the five samples was about 73 °C, 95 °C and 128 °C, respectively. As the microwave heating time increased, the maximum surface temperature increased for all specimens. The specimens containing CGP had the higher temperature compared to CMP-100%. This suggests that the addition of CGP contributes to the improvement of the microwave heating ability.

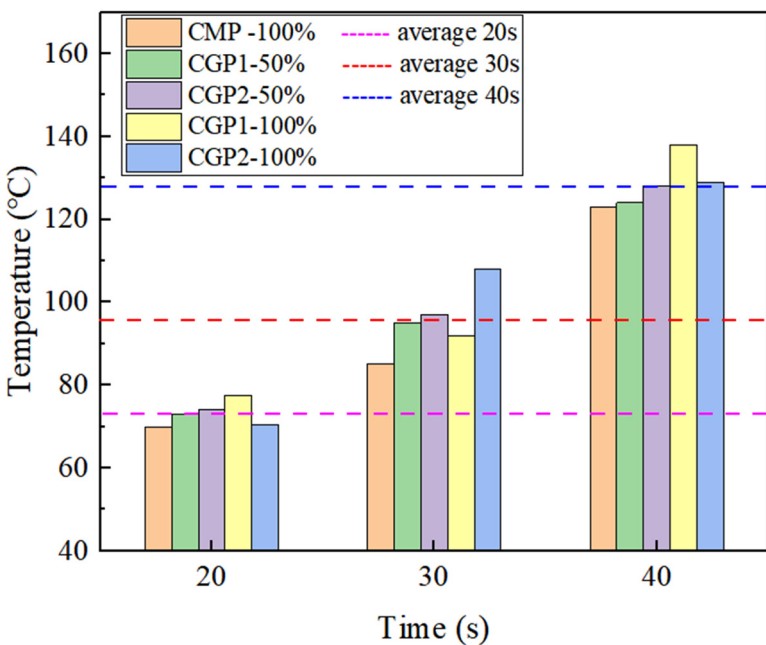

**Figure 10.** Temperature rising trend of various asphalt mixtures.

This may be because the compounds composed of Si and Fe, Al and other related metals have good electromagnetic properties [44,45]. And the comprehensive content of silicon oxide, iron oxide and alumina in CGP is significantly higher than that in CMP (Table 5), which helps to enhance the absorption capacity of microwaves [35,36]. So that CGP has higher microwave absorption efficiency than CMP. According to Table 5, the chemical composition analysis results of CGP show that compared with CGP2, the total content of silicon oxide, iron oxide and alumina in CGP1 is 7.51% higher, but the content of iron oxide is 0.7% lower. Under the same heating time, the maximum temperature of CGP2-50% was generally higher than that of CGP1-50%, which means the iron oxide contributes more to the improvement of microwave heating efficiency of the asphalt mixture. However, this principle is not fit for the asphalt mixture with 100% CGP. As a result, a further investigation is needed to find the relation between filler chemical composition and the microwave heating rate on the asphalt mixture.

Combined with the thermal effect results and based on the average temperature of five groups of specimens under different heating time, a suitable heating time of 30 s was chosen for the comparison of the microwave self-healing ability of different specimens. Because it is the minimum time that allows the asphalt mixture to reach the healing temperature of 85 °C.

### 3.6. Evaluation of Self-Healing Efficiency

Figure 11 shows the fracture energy of the asphalt mixes with different CGP content. With the increase in the number of fracture-healing cycles, the fracture energy decreased gradually for all specimens. After six times of fracture, the fracture energy of CMP-100% was about 495 J/m$^2$, the fracture energy of CGP1-50% was about 638 J/m$^2$, which was approximately 1.29 times that of CMP-100%, and the fracture energy of CGP2-50% was about 642 J/m$^2$, which was approximately 1.30 times that of CMP-100%. In addition, the fracture energy of CGP1-100% and CGP2-100% were about 541 J/m$^2$ and 537 J/m$^2$, respectively, which were still higher than that of CMP-100%. This might be due to the fact that the CGP contained more silicon oxide and metal oxides than traditional mineral powder and generated more heat to achieve higher temperatures for crack closing (as presented in Figure 10).

Figure 12 presents the microwave heating healing rate of the asphalt mixture samples under different fracture cycles. After all six semi-circular bending tests, the healing rates of

the five groups of samples were all above 40%. The CGP1-50% and CGP2-50% samples showed the highest healing rate in the first four bending and healing cycles. With an increasing number of fracture healing cycles, the healing rates of the asphalt mixture containing CGP gradually reduced to a similar level as the control group. This finding indicates that the addition of CGP significantly improved the microwave healing effect of the asphalt mixture in the first four bending and healing cycles, demonstrating the feasibility of using CGP instead of CMP to improve the microwave healing efficiency of asphalt mixtures and benefit the recycling and reutilization of coal gangue. The reason why the healing efficiency of the asphalt mixture containing CGP in the later period of this experiment is lower than that of CMP-100 % may be that the asphalt mixture containing CGP aged more rapidly at long-term high temperature, which reduced its mechanical properties and healing efficiency [46].

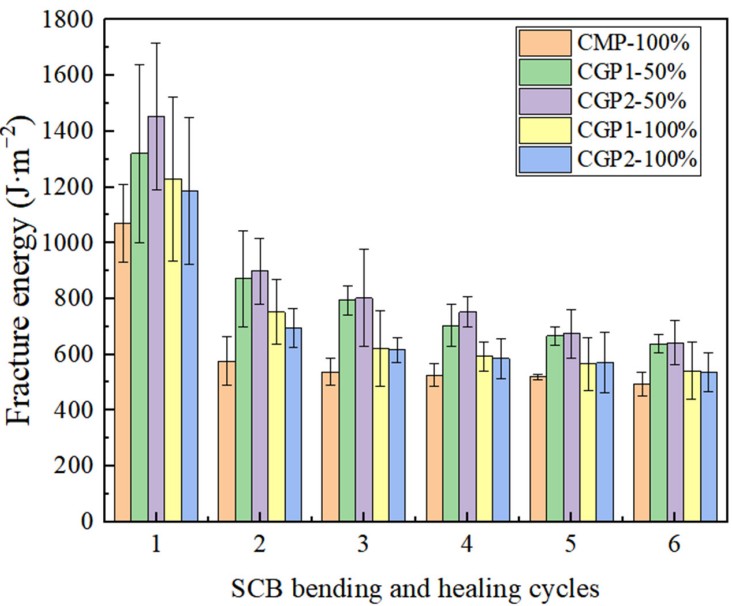

**Figure 11.** Fracture energy of various asphalt mixes.

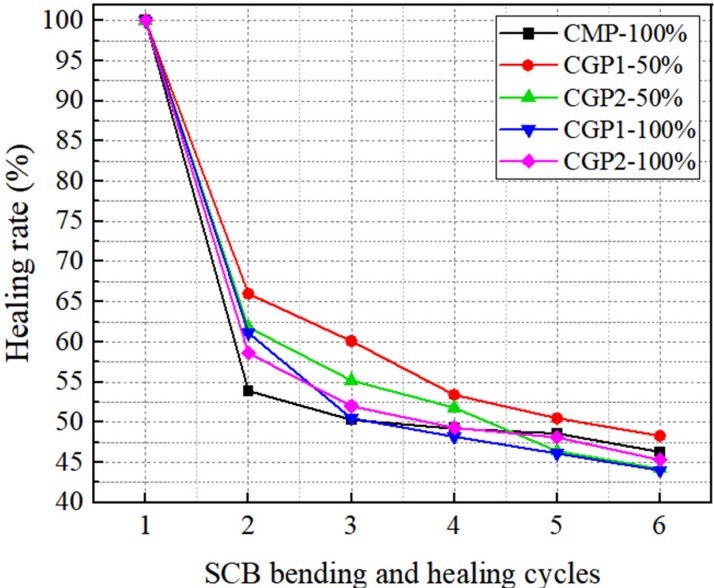

**Figure 12.** Microwave heating healing rate of various asphalt mixes.

## 4. Conclusions

This study investigated the effects of using coal gangue powder (CGP) as a replacement for traditional limestone filler in asphalt mixes, in combination with basalt aggregates. The mechanical performance, water stability, adhesion, low-temperature cracking resistance, microwave heating capacity, and self-healing efficiency of the asphalt mixes were evaluated. The conclusions drawn from the study are as follows:

(1) Residual Marshall stability and scattering losses: The addition of CGP to asphalt mixes improved the residual Marshall stability and reduced the scattering losses. Specifically, CGP2-100% had the highest residual Marshall stability (96.7%), while CGP1-100% had the lowest scattering losses (4.6%). This enhancement in stability is attributed to the increased interaction between asphalt and basalt due to CGP.

(2) Water stability and adhesion: The water stability and adhesion between asphalt and basalt were improved by incorporating CGP into the mixtures. Both CGP types positively affected water stability, with CGP2 showing a slightly greater residual Marshall stability improvement.

(3) Low-temperature crack resistance: The experimental results indicated that the low-temperature crack resistance of asphalt mixtures was enhanced by the addition of CGP, particularly when the CGP content was 50%. This improved crack resistance contributes to the durability of the asphalt pavement.

(4) Microwave heating capacity: The CGP-enhanced asphalt mixes exhibited improved microwave heating speed compared to limestone powder. CGP's higher content of silicon oxide and metal oxides, such as alumina and iron oxide, contributed to this effect. The optimal microwave heating temperature was achieved at 30 s.

(5) Self-healing efficiency: The use of CGP in asphalt mixtures improved the fracture energy and microwave self-healing efficiency, especially during the initial four fracture and healing cycles. This indicates that CGP enhances the asphalt mixture's ability to self-heal, which is crucial for pavement sustainability.

## 5. Future Recommendation

In summary, coal gangue powder as a microwave absorbing material for asphalt mixture has become an important direction to promote the reuse of coal gangue. For the influence of the combined action of coal gangue and basalt, the following problems need to be solved in the future:

(1) This experiment only sets three groups of substitution rates to confirm the feasibility of the combined effect of basalt and coal gangue to improve the microwave heating performance of asphalt mixtures. Therefore, in the future, the best replacement rate of coal gangue should be studied, so that the asphalt mixture can achieve the best healing temperature at the fastest speed.

(2) The influence of the distribution uniformity of coal gangue and basalt in the mixture on the electromagnetic field is still unclear, and the relationship between the chemical composition of the filler and the heating rate should be further studied.

(3) In the future, how to reduce the adverse effects of excessive CGP on the low-temperature crack resistance of asphalt mixture should be studied, so as to further improve the utilization rate of coal gangue.

**Author Contributions:** Conceptualization, S.X., X.G., B.Z. and J.L.; methodology, S.X., X.Y. and Q.T.; validation, S.X., B.Z. and X.Y.; formal analysis, S.X., B.Z. and Q.T.; data curation, B.Z., X.Y. and Q.T.; writing—original draft preparation, B.Z. and J.L.; writing—review and editing, S.X., X.G. and B.Z. All authors have read and agreed to the published version of the manuscript.

**Funding:** This research was supported by the National Natural Science Foundation of China (No. 52108416), Hubei Science and Technology Innovation Talent and Service Project (International Science and Technology Cooperation) (No. 2022EHB006), Key R&D Program of Guangxi Province (No. 2021AB26023), Provincial College Students Innovative Training Project (No. S202310497248) and the Independent Innovation Foundation of Wuhan University of Technology (No. 223131001).

**Institutional Review Board Statement:** Not applicable.

**Informed Consent Statement:** Not applicable.

**Data Availability Statement:** The raw/processed data required to reproduce these findings cannot be shared at this time as the data also forms part of an ongoing study.

**Acknowledgments:** Support from the project is greatly appreciated. The authors also wish to thank the researchers from the State Key Laboratory of Silicate Building Materials.

**Conflicts of Interest:** The authors declare no conflict of interest.

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
