# Peer review of "Microwave Heating Healing of Asphalt Mixture with Coal Gangue Powder and Basalt Aggregate"

_sustainability, doi:10.3390/su151712986_

Round 1
Reviewer 1 Report
1. Page 3, lines 90 to 160 “Material and Methods….”: Can authors provide a more detailed explanation of the experimental methodology used in the study, including sample preparation, testing protocols, and equipment used?
2. Page 9, lines 222-223:
the addition of CGP reduced the standard Marshall strength of the asphalt mixture. How would authors explain this reduction in strength, and were there any trade-offs that needed to be considered in achieving the desired properties?
3. Page 5, lines 131 to 145, How was the Marshall stability test (T0709) conducted? What were the testing conditions, and how many replicates were performed for each mixture?
4. Page 5, lines 125 -129: Can authors provide more details about the XRF detection method used for chemical composition analysis? Specifically, what elements were analyzed, and what equipment and settings were used for the XRF measurements?
5. Page 6 lines 148 to 156, in the Cantabro test, how were the adhesion property and water stability of the asphalt mixture analyzed? Can author elaborate on the procedure and the criteria used to evaluate the scattered losses (∆S)?
6. Page 6, lines 162 - 173: For the low-temperature crack resistance evaluation, how were the semi-circular bending tests conducted? Were any specific parameters or conditions employed during testing?
7. Page 9-, lines 252-253: indicates that CGP can improve the water stability, low-temperature fracture resistance, and microwave heating ability of asphalt mixtures. Can authors elaborate on the underlying mechanisms that lead to these improvements?
8. While the study compared the effects of two types of CGP from different sources, did authors observe any significant variations in the performance of asphalt mixtures with CGP from Taiyuan, Shanxi, and Zhangjiakou, Hebei?
9. Page 10 line 287 – 289 mentions that excessive CGP content might negatively impact low-temperature crack resistance. Can the author provide more details on the observed trends and quantify the extent of the negative impact?
10. Page 12 line 318 to 328, the evaluation of self-healing efficiency, were there any noticeable differences in healing rate between the initial cycles and later cycles? If so, what could be the reason behind these variations?
11. When discussing the microwave heating ability of the asphalt mixtures, were there any specific trends observed between the CGP content and the heating efficiency? For instance, was there an optimal CGP content that resulted in the highest heating efficiency?
12. What future research directions do authors suggest based on the findings of this study, and how can further investigations build upon this work to enhance the understanding and practical applications of CGP and basalt in asphalt mixtures?
Reviewer 2 Report
As comments and directions for further research, the following can be indicated:
1. The frequency and power of the microwave energy source are not justified.
2. The dielectric and magnetic properties of the processed material are not considered.
3. The depth of penetration of the electromagnetic wave and the intensity of heating of the material have not been studied.
4. Energy efficiency for heating has not been evaluated.
5. Conclusions on the work done can be significantly improved.
Round 2
Reviewer 1 Report
This study investigated the effects of using coal gangue powder (CGP) as a replacement for traditional limestone filler in asphalt mixes, in combination with basalt aggregates. The mechanical performance, water stability, adhesion, low-temperature cracking resistance, microwave heating capacity, and self-healing efficiency of the asphalt mixes were evaluated. The conclusions drawn from the study are as follows:
Ø Residual Marshall Stability and Scattering Losses: The addition of CGP to asphalt mixes improved the residual Marshall stability and reduced the scattering losses. Specifically, CGP2-100% had the highest residual Marshall stability (96.7%), while CGP1-100% had the lowest scattering losses (4.6%). This enhancement in stability is attributed to the increased interaction between asphalt and basalt due to CGP.
Ø Water Stability and Adhesion: The water stability and adhesion between asphalt and basalt were improved by incorporating CGP into the mixtures. Both CGP types positively affected water stability, with CGP2 showing slightly better residual Marshall stability improvement.
Ø Low-Temperature Crack Resistance: The experimental results indicated that the low-temperature crack resistance of asphalt mixtures was enhanced by the addition of CGP, particularly when the CGP content was 50%. This improved crack resistance contributes to the durability of the asphalt pavement.
Ø Microwave Heating Capacity: The CGP-enhanced asphalt mixes exhibited improved microwave heating speed compared to limestone powder. CGP's higher content of silicon oxide and metal oxides, such as alumina and iron oxide, contributed to this effect. The optimal microwave heating temperature was achieved at 30 seconds.
Ø Self-Healing Efficiency: The use of CGP in asphalt mixtures improved the fracture energy and microwave self-healing efficiency, especially during the initial four fracture and healing cycles. This indicates that CGP enhances the asphalt mixture's ability to self-heal, which is crucial for pavement sustainability.
Reference should be included:
Crack healing performance of hot mix asphalt containing steel slag by microwaves heating
Construction and Building Materials 20 August 2018- Tam Minh Phan
- Tri Ho Minh Le
Minor revision is required
